# Diagnostic and Prognostic Value of Circulating DNA Fragments in Glioblastoma Multiforme Patients

**DOI:** 10.3390/ijms25084221

**Published:** 2024-04-11

**Authors:** Pawel Jarmuzek, Edyta Wawrzyniak-Gramacka, Barbara Morawin, Anna Tylutka, Agnieszka Zembron-Lacny

**Affiliations:** 1Neurosurgery Center University Hospital, Collegium Medicum University of Zielona Gora, 28 Zyty Str., 65-417 Zielona Gora, Poland; p.jarmuzek@cm.uz.zgora.pl; 2Department of Applied and Clinical Physiology, Collegium Medicum University of Zielona Gora, 28 Zyty Str., 65-417 Zielona Gora, Poland; e.gramacka@cm.uz.zgora.pl (E.W.-G.); b.morawin@cm.uz.zgora.pl (B.M.); a.tylutka@cm.uz.zgora.pl (A.T.)

**Keywords:** biomarker, cell free DNA, inflammation, overall survival time, tumour-derived DNA

## Abstract

Novel blood-circulating molecules, as potential biomarkers for glioblastoma multiforme (GBM) diagnosis and monitoring, are attracting particular attention due to limitations of imaging modalities and invasive tissue biopsy procedures. This study aims to assess the diagnostic and prognostic values of circulating cell-free DNA (cfDNA) in relation to inflammatory status in GBM patients and to determine the concentration and average size of DNA fragments typical of tumour-derived DNA fractions. Preoperative plasma samples from 40 patients (GBM 65.0 ± 11.3 years) and 40 healthy controls (HC 70.4 ± 5.4 years) were compared. The cfDNA concentrations and lengths were measured using the electrophoresis platform, and inflammatory indices (NLR, PLR, LMR, and SII) were calculated from complete blood cell analysis. More fragmented cfDNA and 4-fold higher 50–700 bp cfDNA concentrations were detected in GBM patients than in healthy controls. The average cfDNA size in the GBM group was significantly longer (median 336 bp) than in the HC group (median 271 bp). Optimal threshold values were 1265 pg/μL for 50–700 bp cfDNA (AUC = 0.857) and 290 bp for average cfDNA size (AUC = 0.814). A Kaplan–Meier survival curves analysis also demonstrated a higher mortality risk in the GBM group with a cut-off >303 bp cfDNA. This study is the first to have revealed glioblastoma association with high levels of cfDNA > 1000 pg/μL of 50–700 bp in length, which can be aggravated by immunoinflammatory reactivity.

## 1. Introduction

Glioblastoma (GBM) is the most common malignant primary brain tumour in adults and follows an aggressive clinical course associated with a median survival of 217 days and a median age of 63 years for the Polish population. With regard to age, the survival probability decreased by 50% in patients aged 63 years [1]. Therefore, age is one of the most important prognostic factors in the treatment of glioblastoma [2,3]. The life expectancy for Poland in 2024 has increased by 0.21% compared to 2023 and currently amounts to 79.43 years, whereas during the last half century it has risen by more than 10 years for both men and women [4]. This trend is similar worldwide and, accordingly, the concept of the “elderly” is changing. Based on the analysis of various data on physical and psychological health and the recorded increases in surgical risk factors and hospital mortality, several studies have suggested that the age of 65–74 years should be considered “pre-old age” and 75 years or older should be regarded as “old age”, [5,6]. Approximately 90% of GBM occur in older patients, whereas in younger patients, it usually develops from a lower-grade glioma. Our retrospective study showed that the survival probability decreases considerably faster in older (63–90 years) than young patients (23–63 years) with a high-grade glioma [1].

Current practices involve assessing molecular markers in tumour biopsies, which necessitate invasive neurosurgical procedures and pose risks to patients. In glioblastoma, diagnosis and monitoring typically rely on imaging techniques, which are unable to discriminate between tumour progression and pseudoprogression, which is defined as some treatment-related changes that mimic tumour progression [7]. This can lead to misinterpretation of therapy response and can delay clinical interventions. Therefore, the assessment of molecular markers at the tumour tissue is necessary for GBM diagnosis and predictions of patients’ prognosis and their response to treatment [8,9,10]. Alas, neurosurgical procedures for obtaining tumour tissue are so invasive and complication-prone that they can only be performed for patients in a good general condition, with tumours located in noncritical parts of the brain. Furthermore, GBM heterogenicity causes tissue specimens from a single portion of GBM tumour to be poorly representative of the whole tumour condition [11]. Also, real-time assessment of tumour tissue dynamics via multiple biopsies throughout treatment cannot often be performed, due to their high levels of invasiveness and risk for the patient. Therefore, identification of new tumour biomarkers in the bloodstream has gained increasing attention, as the advantages of such a parameter’s evaluation include easy access, low cost, and minimal invasiveness [7].

Some studies have highlighted potential benefits of the measurement of cell-free DNA fragments (cfDNA), which are released from the tumour and healthy cells into the bloodstream as a result of apoptosis, necrosis, NETosis, and active secretion via extracellular vesicles [9,12,13,14,15,16]. According to the American Society of Clinical Oncology and the College of American Pathologists, the largest fraction of cfDNA in patients with cancer is derived from the tumour tissue of origin [17]. The extracellular DNA pool includes total circulating cell-free DNA (cfDNA), circulating tumour-derived cell free DNA (ctDNA), and circulating mitochondrial DNA (mtDNA). Under normal conditions, cfDNA concentration is typically low (within 100 pg/μL), but it increases significantly under pathological conditions, including cancer > 1000 pg/μL [18]. In total, 85% of plasma cfDNA fragments in cancer patients are 166 base pairs (bp), 10% are 332 bp, and 5% are 498 bp in length. Larger cell-free DNA fragments, i.e., ~10,000 bp in length, are the products of necrosis, whereas DNA fragments shorter than 1000 bp, particularly of 180 bp or multiples of this size, are reminiscent of the oligonucleosomal DNA ladder observed in apoptotic cells [12]. Thierry and Sanchez [19] defined a list of parametrics to distinguish cancer vs. healthy individual circulating cfDNA extracts. Overall, cancer patient cfDNA shows subtle but reliable differences relative to healthy subjects. In individuals with cancer, cfDNA differences include a higher number of fragments below 150 bp, as well as a lower number of fragments between 151 and 218 bp; with dinucleosome-corresponding fragments peaking at 300 vs. 330 bp [20,21]. The elevated levels of total cfDNA have been demonstrated in malignant tumours among adults, including glioblastoma patients, relative to patients with non-neoplastic diseases [12]. However, cfDNA levels in brain tumours are reduced by 60% in medulloblastoma and by 90% in low-grade glioma, as compared to systemic malignancies [12,13,14]. In addition, the length of the DNA fragments and their source in glioblastoma are unknown and the measurement of cfDNA in GBM patients for clinical applications remains a multifaceted problem. Currently, circulating cfDNA and ctDNA fractions are analysed in the context of inflammation, as they appear to be promising potential biomarkers for the early diagnosis or prognosis in glioblastoma [22,23,24]. To date, only two meta-analyses have been reported by MacMahon et al. [13] and Jarmuzek et al. [25]. Both studies showed that cfDNA appeared to be a significantly sensitive and specific biomarker in adults with low- and high-grade gliomas; however, further studies should be conducted with glioblastoma as a target. Therefore, this study compared the preoperative status of circulating cfDNA in GBM patients and healthy controls to assess the diagnostic and prognostic values of cfDNA in relation to inflammatory status and to determine the concentration and average size of cfDNA typical of the tumour-derived DNA fraction.

## 2. Results

### 2.1. Study Population

Among our 40 study patients, 66% were females aged 67.3 ± 9.8 years, with a median survival of 122 days (4–369 days) and 34% were males aged 62.5 ± 12.7 years, with a median survival of 250 days (33–483 days). Significant differences (*p* < 0.01) in average survival time between sexes (females 138 ± 105 days, males 250 ± 140 days) were observed. GBM was mostly located in the supratentorial region (frontal, temporal, and parietal lobes), with the highest incidence detected in the frontal lobe (32.5%). Ki-67 ≥ 30% was recorded at 65%.

### 2.2. Study White Blood Cell Count-Derived Inflammation Indices

White blood cells (WBCs) were found to fall within the referential ranges and were significantly higher in the GBM group than in the HC group. The counts of neutrophils were elevated (*p* < 0.001), whereas lymphocytes tended to reach low values in GBM (*p* = 0.059). The monocyte and platelet counts did not differ between groups. Among the investigated immune cells, neutrophils exceeded reference values and showed the greatest changes reflected by alternations in neutrophil/lymphocyte ratio (NLR) and systemic immune inflammation index (SII) (Table 1). NLR and SII exceeded the reference values, whereas the lymphocyte/monocyte ratio (LMR) and the platelet/lymphocyte ratio (PLR) were found to fall within the referential ranges proposed by Luo et al. [26], but all WBC-derived inflammation indices were elevated in the GBM group compared to the HC group. There is a concurrent rise in NLR observed alongside elevated levels of 50–700 bp cfDNA in glioma patients, which suggests that tumour-associated inflammation appears to be the major contributor to the release of cfDNA into the bloodstream. The results of the receiver operating characteristic (ROC) curves analysis of NLR and SII were >0.8, indicating a potential diagnostic value for clinical prognosis for patients with high-grade glioma (Figure 1A,D), as was the case in our earlier observations [27]. The optimal threshold values corresponded to 2.295 for NLR (AUC = 0.873, specificity 80%, sensitivity 90%, RR = 4.889, 95%CI 0.782–0.963), 3.310 for LMR (AUC = 0.679, specificity 67.5%, sensitivity 65%, RR = 1.976, 95%CI 0.554–0.804), 142 for PLR (AUC = 0.581, specificity 52.5%, sensitivity 72.5%, RR = 1.658, 95%CI 0.444–0.718), and 482 for SII (AUC = 0.809, specificity 72.5%, sensitivity 85%, RR = 3.390, 95%CI 0.705–0.913) (Figure 1A–D).

### 2.3. DNA Analysis

GBM patients displayed a 5-fold higher concentration of total circulating cfDNA (Table 2) which was more scattered than in the HC group (Figure 2). No significant differences in cfDNA level were observed between females and males. The majority of the GBM patients (77%) demonstrated a very high concentration of cfDNA > 2000 pg/μL, which may aggravate immunoinflammatory reactivity, according to Jylhävä et al. [28]. In the HC group, cfDNA > 1000 pg/μL occurred in 47%, mainly in the oldest ones. The cfDNA of 50–700 bp in length was 4-fold higher, but their percentage in the total cfDNA pool was lower in GBM samples. The 50–700 bp fragments constituted ~74% of circulating total cfDNA in GBM samples (Table 2), which means the remaining 26% of cfDNA could be fragments >700 bp, originating from necrosis. The average cfDNA size was longer and the size distribution was more scattered in the GBM samples than in the HC samples, as was the case with cfDNA concentration (Figure 2 and Figure 3). The results of the ROC analysis of total cfDNA, 50–700 bp cfDNA, and average cfDNA size were >0.8, indicating a potential diagnostic value for clinical prognosis for patients with high-grade glioma (Figure 4). The optimal threshold values corresponded to 1561 g/μL for total cfDNA (AUC = 0.891, sensitivity 77.5%, specificity 85%, RR = 4.876, 95%CI 0.823–0.960), 1265 pg/μL for 50–700 bp cfDNA (AUC = 0.857, sensitivity 72.5%, specificity 82.5%, RR = 3.857, 95%CI 0.777–0.938), 82.32% for cfDNA percentage (AUC = 0.736, sensitivity 85%, specificity 57.5%, RR = 2.379, 95%CI 0.624–0.848), and 290 bp for average cfDNA size (AUC = 0.814, specificity 80%, sensitivity 77.5%, RR = 3.805, 95%CI 0.711–0.917) (Figure 4A–D). The highest RR was observed for total cfDNA concentration (RR = 4.876), which means an approx. 5-fold higher probability of GBM development once the cut-off value for cfDNA has been exceeded. Moreover, the highest sensitivity (77.5%) and specificity (85%) were observed for cfDNA, which, in turn, indicates a low level of false positive results during diagnostic procedures using cfDNA. The results of the Kaplan–Meier survival curves analysis, although insignificant, showed a higher mortality risk in GBM patients with 303–416 bp cfDNA than patients with 201–303 bp cfDNA fragments (Figure 5). Kaplan–Meier curves were initially parallel but crossed in the further observation period, in which case, the condition of proportionality of hazards was not met and significant differences between groups could not be detected. The survival times remained higher in patients with 201–303 bp cfDNA size. The factor disturbing the course of the curves was probably the effectivity of the surgical procedure (total or subtotal tumour resection), which reversed the relation cfDNA size and survival times. According to the meta-analysis of Han et al. [29], total or subtotal resection significantly impact survival times in glioblastoma. It should be emphasized that full assessment of the prognostic value of 50–700 bp fragments requires further study to identify the confounding factor on a larger sample size.

## 3. Discussion

Cell death is a complex process which is influenced by a range of factors through a variety of underlying mechanisms, thereby making it difficult to estimate the contribution of different cell death types to the cfDNA pool. While cell death mechanisms are associated with distinct morphological-, biochemical-, and immune-related changes, these processes are molecularly interconnected [30,31]. For example, NETosis is the cell death type that is associated with the formation of neutrophil extracellular traps, potentially induced by damage-associated molecular patterns and more frequently observed in highly inflammatory cancers. Inflammation and, according to recent reports, also NETosis have been implicated as critical factors in carcinogenesis and tumour progression [15,31].

Neutrophils are the first cells to infiltrate under the direction of inflammatory mediators, which is followed by lymphocytes’ and macrophages’ stimulation and recruitment to the site of inflammation. The immune cells within the tumour neutrophil microenvironment are highly plastic and they continuously change their secretory phenotypic and functional characteristics [32]. Strong perturbation of tissue homeostasis leads to the recruitment of immune cells from bone marrow and secondary lymphoid cells, which results in an increase in the counts of neutrophils, monocytes, and monocyte-derived cells in the peripheral blood [32]. We previously observed that among immune cells, neutrophils showed the most considerable changes, especially in patients with Grade 3 and Grade 4 tumours, whereas lymphocytes, monocytes, and platelets did not exhibit significant changes compared to reference levels or the Grade 1 group aged ≤30 years [1]. However, diagnosis and prognosis of younger patients is difficult due to low numbers. In our retrospective study, younger patients constituted a group of only 11 individuals among 358 patients with newly diagnosed glioma who had undergone an operation at the Neurosurgery Centre University Hospital [1]. Approximately 90% of GBM primarily occur in older patients, whereas in younger patients, it usually develops from a lower-grade glioma [1,33,34].

This study consistently showed that the neutrophil count exceeded the reference values and reached several times higher levels in GBM, while other immune cells did not differ between GBM patients and healthy individuals. The neutrophilia inhibits the immune system response by suppressing the cytolytic activity of cytotoxic T cells and natural killer cells, which is expressed in changes in the neutrophil-to-lymphocyte ratio [32]. NLR, as an effective prognostic marker in GBM patients, was demonstrated in our recently reported retrospective study and meta-analysis [1,27]. The Cox model analysis showed that an NLR ≥ 4.56 significantly increased the risk of death in GBM. Among inflammatory indices, NLR demonstrated the greatest impact on the survival time (HR = 1.56, 95%CI 1.145–2.127, *p* = 0.005) [1]. The current study in the GBM vs. HC setting confirmed the diagnostic value of NLR (Figure 1A) and also revealed that tumour-associated inflammation appeared to be the major contributor to the release of cfDNA into the bloodstream. Recently, cfDNA has been regarded as a damage-associated molecular pattern molecule (DAMP) that can initiate an inflammatory response in various diseases, including highly inflammatory cancers [15,30].

Currently conducted cfDNA-based early detection studies focus on extracranial cancers and the early detection of intracranial tumours such as glioma remains challenging, due to the blood–brain barrier for cfDNA transmission into peripheral plasma. The unique architecture of the brain causes the level of detectable cfDNA in brain tumours to be reduced by 60% and by 90% in medulloblastoma and in low-grade glioma, respectively, as compared to various systemic malignancies [35]. There are no available data on high-grade glioma, which indicates that cfDNA detection in glioma patients for clinically relevant purposes remains a challenging and complex problem. Some comparative studies demonstrated that glioma shed the least cfDNA into circulation compared with other cancers, even in patients with advanced disease [36]. We measured 50–700 bp fragments, which constitute the largest fraction of cfDNA derived from the tumour tissue of origin. Their average size was longer (201–416 bp) and the concentration was 4-fold higher in GBM samples than in HC samples. Circulating tumour-derived DNA, the subset of cfDNA specifically shed from tumour cells, has recently become a promising biomarker in patients with advanced cancer. The meta-analysis performed by MacMahon et al. [13] revealed an improved biomarker performance for cerebrospinal fluid (AUC = 0.947) vs. plasma (AUC = 0.741) ctDNA, although this did not reach statistical significance. Our meta-analysis of two studies comprising 104 patients demonstrated a worse prognosis in patients with a higher cfDNA (HR 2.35, 95%CI 1.27–4.36, *p* < 0.01). One of the analysed studies reported a significant difference between overall cfDNA burden in patients with glioblastoma vs. healthy controls and a worse prognosis in patients with glioblastoma with higher total cfDNA concentrations [37].

In cancer patients, cfDNA fragments released by apoptotic and/or necrotic cells typically reached a total amount of less than 0.1–5% of total cell-free DNA, which correlated with tumour type, grade, and burden [38]. The present study demonstrated a 5-fold higher concentration of total cfDNA and its more fragmented pattern in GBM patients than in the HC group (Figure 2). Most of our study patients with glioblastoma demonstrated very high concentrations of cfDNA, above 2000 pg/μL, which was associated with increased neutrophils counts and NLR. A Kaplan–Meier survival curves analysis indicated a higher mortality risk in GBM patients with 303–416 bp cfDNA than in patients with 201–303 bp. Therefore, we suggest that 201–303 bp cfDNA could be considered a potential marker of therapeutic decision and effectiveness of adjuvant treatment, i.e., chemotherapy and radiotherapy. We also expect that cfDNA levels may be helpful in differentiating and evaluating MRI follow-up postoperative lesions (pseudoprogression) from actual tumour regrowth (progression), yet more research on a larger population of patients to substantiate cfDNA potential and to standardize this assay. 

Fragments of 50–700 bp in length constituted 74.38% of total cfDNA in GBM samples and 86.87% in HC samples, which reflects the filtration effect of the blood–brain barrier in glioblastoma. It is highly probable that the shorter cfDNA < 50 bp could originate from cut apoptotic-derived 140 bp–180 bp fragments, but short cfDNA are more frequent in patients with metastatic cancer [39]. Therefore, we postulate that cfDNA could be potential products of NETosis associated with activated neutrophils present in the blood periphery. In healthy individuals, Moss et al. [40] estimated that 55% of circulating cfDNA was derived from white blood cells, 30% from erythrocyte progenitors, and 10% from vascular endothelial cells, accounting for 95% of cfDNA. In turn, Pastor et al. [15] detected NETs in the pathological conditions, in which high concentrations of circulating cfDNA were reported, such as autoimmune diseases, inflammatory diseases, sepsis, thrombotic illnesses, and cancer. The higher neutrophil counts and their longer lifespan (activated in cancers) in patients with cancer could increase the formation of NETs and NETs’ by-products [15]. NETosis plays a crucial role in circulating cfDNA production, notably in the degradation of NETs-derived web-like chromatin in blood. Our earlier observations revealed that the release of cfDNA into the circulation was proportional to the counts of white blood cells and the severity of the systemic inflammation in older adults [41]. Therefore, our assumption is that cfDNA profiling could be used not only as a diagnostic and prognostic marker for patients with glioblastoma, but also as a predictor of immune status in a population at high risk of cancer development. 

Based on the outcomes of this study, our conclusion is that glioblastoma promotes an immune-inflammatory response which disrupts the blood–brain barrier and leads to the cfDNA transmission into peripheral blood. Circulating cell-free DNA released by both tumour cells and activated neutrophils may provide highly specific markers for the detection and prognosis of glioblastoma. To the best of our knowledge, this is the first study to have revealed that glioblastoma is associated with high levels of cfDNA > 1000 pg/μL of 50–700 bp in length, which can be aggravated by immunoinflammatory reactivity. Furthermore, the ROC analysis of 50–700 bp cfDNA concentration indicated a potential diagnostic value for clinical prognosis for patients with GBM.

## 4. Materials and Methods

### 4.1. Study Population

The study was carried out in forty patients aged 65.0 ± 11.3 years (females *n =* 24, males *n* = 16) with newly diagnosed glioblastoma (GBM), who had undergone an operation at the Neurosurgery Centre University Hospital, Zielona Gora (Table 3). There were no extracranial metastases recorded in newly diagnosed glioblastoma during the follow-up period. The pathological diagnoses were based on the classification of CNS tumours [42]. The overall survival was defined as the interval between the diagnosis and death. For the patients who had not died prior to the last follow-up, the overall survival was censored at the date of the last follow-up. All patients underwent a craniotomy on GBM, with either total or subtotal resection. The following exclusion criteria were used: biopsy only, age below 18 years, no definite diagnosis, incomplete baseline clinical data, adjuvant therapy like chemotherapy or radiotherapy received before operation, malnutrition, and perioperative mortality (survival < 20 days). Importantly, every patient diagnosed with a primary brain tumour and enrolled in our study had been diagnosed only recently and no prior specific treatment, including glucocorticoids, was applied. The GBM group was compared to forty healthy individuals aged 70.4 ± 5.4 years (females *n* = 28, males *n* = 12) recruited from the University of the Third Age in Zielona Gora (Poland), which is an organization which encourages adults over 60 years of age to stay active by participating in many educational programmes. The current health status of the control group was assessed on the basis of medical records at a routine follow-up visit to a primary care physician. On the basis of the medical interview, the following exclusion criteria were applied: acute infectious and autoimmunological diseases, uncontrolled hypertension and/or diabetes, oncologic diseases, and neurodegenerative diseases. The study protocol was approved by the Bioethics Commission at the University of Zielona Gora, Poland (No. 16/2022), in accordance with the Helsinki Declaration.

### 4.2. Clinical Assessment

Medical records were reviewed and the following clinical data were collected: gender, age at operation, locations and hemisphere of tumours, pathological diagnoses, and some biomarkers. The Ki-67 proliferation index was expressed as the percentage of cells with Ki-67-positive immunostained nuclei using the Ventana BenchMark GX (Ventana Medical Systems Inc., Tucson, AZ, USA). The expression of Ki-67 was categorised into the following two groups: low and intermediate (Ki-67 < 30%) and high (Ki-67 ≥ 30%), according to recommendations by Chen et al. [43]. The data on postoperative adjuvant therapies and survival time were collected through documentation analysis.

### 4.3. Blood Samples Collection

The blood samples of patients with GBM were collected from the median cubital vein in the morning between 7.00 and 9.00 on the day of admission to hospital (one day before the surgical resection) and before the introduction of steroid therapy. Blood samples were collected in S-Monovette-EDTA K2 anticoagulant tubes (Sarstedt AG & Co. KG, Nümbrecht, Germany) and immediately subjected to haematology analysis in the laboratory of the University Hospital in Zielona Gora. For further experiments, blood samples were centrifuged at 3000 rpm for 10 min and aliquots of plasma were stored at −80 °C. 

### 4.4. White Blood Cell Count-Derived Inflammation Indices

Haematological parameters including WBCs, platelet count, and differential WBCs were determined using Sysmex XN-1000 (Sysmex Europe Gmbh, Norderstedt, Germany). The neutrophil-to-lymphocyte ratio (NLR, the platelet-to-lymphocyte ratio (PLR), the lymphocyte-to-monocyte ratio (LMR), and the systemic immune inflammation index (SII = (platelets × neutrophils)/lymphocytes)) were calculated and compared to reference values, according to Luo et al. [26].

### 4.5. DNA Analysis

The total circulating cell-free DNA fragments were isolated using the Cell-Free AX DNA Mini Kit, in accordance with the manufacturer’s protocol (A&A Biotechnology, Gdansk, Poland). The cell-free AX Kit is dedicated to the isolation of cell-free DNA from serum/plasma (sample volume 100 μL) and circulating tumour DNA (ctDNA). Due to the high quality of isolated DNA, it can be used for a wide spectrum of applications, e.g., the detection and monitoring of cancer. Then, the cell-free DNA Screen Tape Kit (Agilent Technologies, Inc. Headquarters, Santa Clara, CA, USA) was used to determine accurate length (from 50 bp to 700 bp) and concentration of cfDNA at fully automated electrophoresis Cell-Agilent 4150 Tape Station (Agilent Technologies, Inc. Headquarters, Santa Clara, CA, USA).

### 4.6. Statistical Analysis

Statistical analyses and figures were performed using R 4.2.1 software [44]. The variables were reported as mean values ± standard deviation (SD) and median (Med). The assumptions for the use of parametric or nonparametric tests were checked using the Shapiro–Wilk and Levene’s tests to assess the normality of the distributions and the homogeneity of variances, respectively. The significant differences in mean values between the GBM and HC groups were evaluated using a one-way ANOVA. If the normality and homogeneity assumptions were violated, the Mann–Whitney nonparametric test was used. The predictive value of inflammatory variables and cfDNA fragments were evaluated using the receiver operating characteristic curve (ROC). The area under the ROC curve (AUC) and confidence interval (95%CI) were used to provide an aggregate measure of performance across all possible classification thresholds. The optimal threshold value for clinical stratification (cut-off value) was obtained by calculating the Youden index. Relative risk (RR) was calculated to demonstrate the risk for an event for the GBM group to the risks for the HC group. Survival curves were plotted using the Kaplan–Meier method and were compared using the log-rank test. Statistical significance was set at *p* < 0.05.

## Figures and Tables

**Figure 1 ijms-25-04221-f001:**
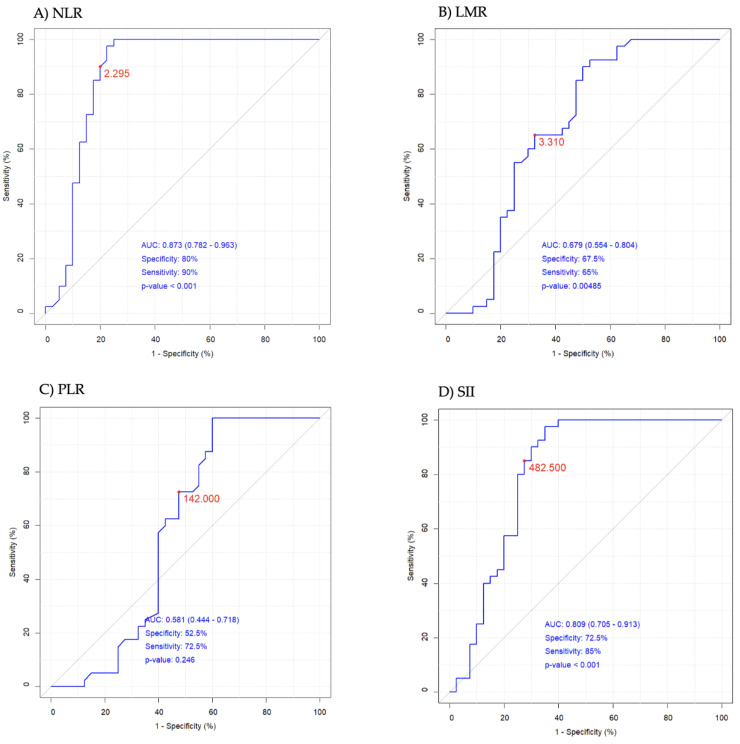
The receiver operating characteristic (ROC) curves of (**A**) NLR—neutrophil/lymphocyte ratio, (**B**) LMR—lymphocyte/monocyte ratio, (**C**) PLR—platelets/lymphocytes ratio, and (**D**) SII—systemic immune inflammation index. The red dot on each plot represents the optimal threshold value (cut-off). AUC represents area under the ROC curve.

**Figure 2 ijms-25-04221-f002:**
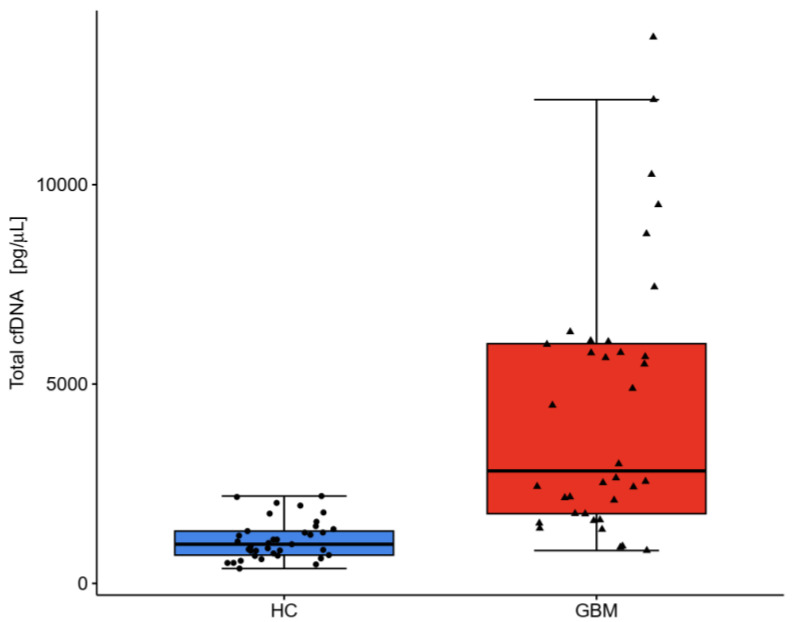
Concentration distribution of circulating cell-free DNA (total cfDNA) fragments in healthy controls (HC min 373 pg/μL, max 3960 pg/μL) and glioblastoma patients (GBM min 825 pg/μL, max 32,547 pg/μL).

**Figure 3 ijms-25-04221-f003:**
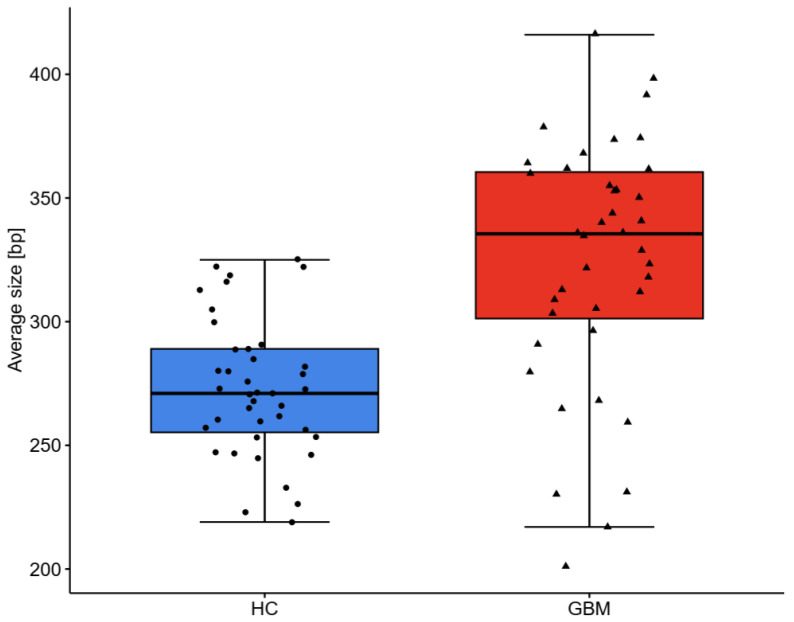
Size distribution of circulating cfDNA fragments in healthy controls (HC min 219 bp, max 325 bp) and glioblastoma patients (GBM min 201 bp, max 416 bp).

**Figure 4 ijms-25-04221-f004:**
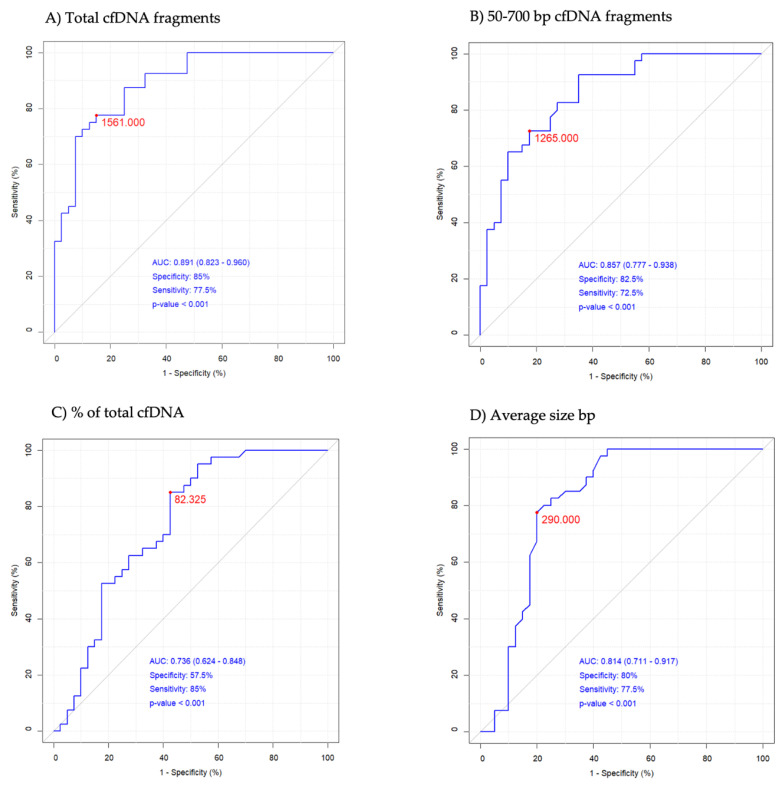
The receiver operating characteristic (ROC) curves of circulating cell-free DNA (cfDNA). The red dot on each plot represents the optimal threshold value (cut-off). AUC represents area under the ROC curve.

**Figure 5 ijms-25-04221-f005:**
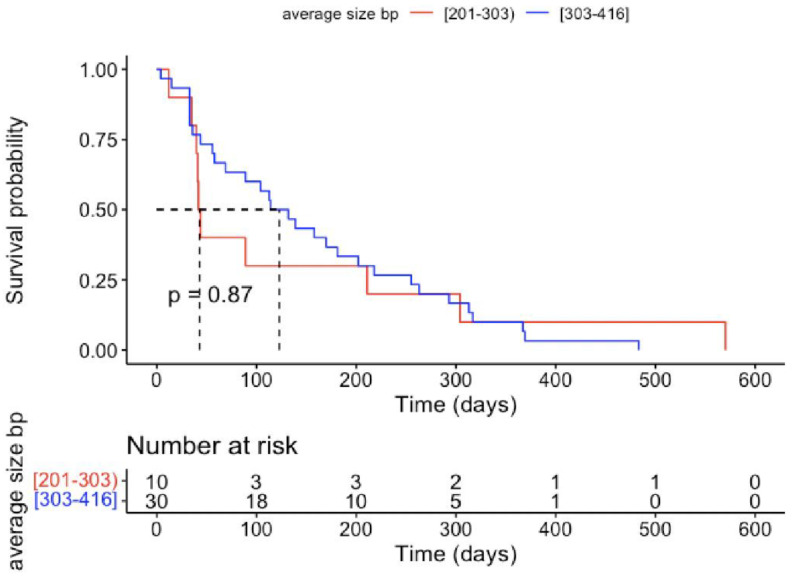
Kaplan–Meier survival curves of GBM patients with different cut-off values of average cfDNA size.

**Table 1 ijms-25-04221-t001:** White blood cell count-derived inflammation indices.

Variables	Reference Values	HC *n* = 40		GBM *n* = 40		
		Mean ± SD	Med (IQR 25–75%)	Mean ± SD	Med (IQR 25–75%)	HC vs. GBM *p* Level
WBCs (10^3^/µL)	4.0–10.2	5.73 ± 1.55	5.50 (4.95–6.34)	8.47 ± 3.09	7.87 (5.83–10.33)	<0.001
Neutrophils (10^3^/µL)	2.0–6.9	2.99 ± 1.06	2.88 (2.42–3.39)	10.69 ± 3.09	7.87 (5.85–12.79)	<0.001
Lymphocytes (10^3^/µL)	0.6–3.4	1.94 ± 0.58	1.85 (1.68–2.15)	1.83 ± 1.36	1.50 (0.87–2.34)	0.059
Monocytes (10^3^/µL)	0.00–0.90	0.52 ± 0.12	0.53 (0.46–0.58)	0.73 ± 0.78	0.56 (0.37–0.79)	0.695
Platelets (10^3^/µL)	140–420	227 ± 53	223 (198–247)	216 ± 111	206 (136–244)	0.373
NLR	0.87–4.15	1.58 ± 0.50	1.55 (1.33–1.84)	8.33 ± 7.30	5.71 (3.00–11.81)	<0.001
PLR	47–198	123 ± 34	115 (103–154)	166 ± 112	146 (87–207)	0.085
LMR	2.45–8.77	3.81 ± 1.08	3.69 (2.99–4.56)	4.41 ± 6.44	2.69 (1.75–3.71)	<0.001
SII	142–808	366 ± 163	338 (260–452)	1796 ± 1704	1243 (447–2558)	<0.001

Abbreviations: HC—healthy control, GBM—glioblastoma, SD—standard deviation, Med—median, IQR—interquartile range, WBCs—white blood cells, NLR—neutrophil/lymphocyte ratio, PLR—platelet/lymphocyte ratio, LMR—lymphocyte/monocyte ratio, SII—systemic immune inflammation index. The last column shows the *p*-values of the one-way ANOVA or the Mann–Whitney nonparametric test (if the normality assumption is violated).

**Table 2 ijms-25-04221-t002:** The concentration of cell-free DNA (cfDNA) in healthy control and glioblastoma patients.

Variables	HC *n* = 40		GBM *n* = 40		HC vs. GBM *p* Level
	Mean ± SD	Med (IQR 25–75%)	Mean ± SD	Med (IQR 25–75%)	
Total cfDNA pg/μL	1290 ± 851	1032 (741–1463)	6520 ± 7374	4678 (2003–6590)	<0.001
50–700 bp cfDNA pg/μL	1131 ± 762	870 (649–1375)	4360 ± 4371	2435 (1515–5312)	<0.001
% of total cfDNA	86.87 ± 5.39	87.79 (83.30–91.00)	74.38 ± 17.92	80.32 (67.59–86.60)	<0.001
Average size bp	272 ± 27	271 (255–289)	324 ± 50	336 (301–360)	<0.001

Abbreviations: HC—healthy control, GBM—glioblastoma, SD—standard deviation, Med—median, IQR—interquartile range. The last column shows the *p*-values of the one-way ANOVA or the Mann–Whitney nonparametric test (if the normality assumption is violated).

**Table 3 ijms-25-04221-t003:** The clinical characteristics of patients with glial tumours (*n* = 40).

		Value
Follow-up period	Mean ± SD (day)	175.7 ± 133.2
	Median (range)	149.5 (4–570)
Age at operation	Mean ± SD (year)	65.0 ± 11.3
	Median (range)	63.6 (38–89)
Gender	Females	24 (60%)
	Males	16 (40%)
Hemisphere	Left	17 (42.5%)
	Right	20 (50%)
	Midline or bilateral	3 (7.5%)
Location	Frontal lobe	13 (32.5%)
	Temporal lobe	9 (22.5%)
	Parietal lobe	7 (17.5%)
	Occipital lobe	3 (7.5%)
	Subtentorial location	2 (5%)
	Multifocal	1 (1.5%)
Ki-67	≥30%	26 (65%)
	<30%	14 (35%)

Abbreviations: Ki-67, a nuclear protein and a key marker associated with proliferating cancer cells.

## Data Availability

The raw data supporting the conclusions of this article will be made available by the authors, without undue reservation.

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
