# Peer review of "Diagnostic and Prognostic Value of Circulating DNA Fragments in Glioblastoma Multiforme Patients"

_ijms, 2024, doi:10.3390/ijms25084221_

Round 1

Reviewer 1 Report

Comments and Suggestions for Authors

This study evaluates the use of circulating cell-free DNA to diagnose and predict the course of glioblastoma in elderly patients. The study contains interesting results and observations, however, the authors need to answer a number of questions before the manuscript is accepted for publication:
1. the authors need to explain briefly (1-2 sentences) at the section 2.2 the sense of evaluation of NLR, PLR and LMR in the context of circulating DNA.
2. all ratios (NLR, PLR, LMR) cannot have dimensionality. (1.5*10^4 cells/ul) / (1*10^3 cells/ul) = 15 units (not cells/ul). Need to be corrected throughout the text.
3. materials and methods:
- need to describe how CtDNA has been identified in the pool of CfDNA. How to distinguish ctDNA from cfDNA?
- the authors describe in detail how the normality of a distribution can be determined, but do not specify how the data of a non-normal distribution are described. They should be described as the median(25%;75%) but not as the mean/median ± standard deviation. If you have data with non-normal distributions in your study or not, please, adjust the description of Statistical analysis.
4. Discussion:
- if CtDNA is 1% of CfDNA or less, what kind of DNA underlies the 4-fold increment of CfDNA in GBM. What is the source of this CfDNA - need to be discussed;
- need to discuss the possibility of the described approach for:
--- diagnosis and prognosis of GBM in younger patients;
--- diagnosis of progression and pseudoprogression of GBM (ability to differentiate);
--- predicting the efficacy of chemotherapy and radiotherapy.
- the authors entitled their article as "Diagnostic and prognostic value of circulating DNA fragments ...", however, they did not make any clear conclusions about the possible application of the proposed approach for the diagnosis and prognosis of GBM outcome. In the penultimate paragraph of the discussion, it is necessary to summarize the possible applications of the proposed approach and all its limitations. One of the most significant limitations of this approach is its nonspecificity: the amount of CfDNA can increase in other pathologies (cancers, infections and autoimmune disorders). It should be emphasized that evaluation of CfDNA/CtDNA levels and/or spectrum is only appropriate when glioblastoma has already been diagnosed by MRI or other clinically approved approach.
5. Institutional Review Board Statement and Informed Consent Statement should be corrected to meet the requirements of IJMS. The study includes samples obtained from humans!!!

Author Response

Review 1

We greatly appreciate your time and effort dedicated to providing feedback on our manuscript and we are grateful for the insightful comments on and valuable improvements to our paper. All the suggestions helped us to evaluate our outcomes even more precisely in order to deliver improved, high quality scientific manuscript which we hope will now meet the high standards of International Journal of Molecular Sciences.

Comments and Suggestions for Authors

This study evaluates the use of circulating cell-free DNA to diagnose and predict the course of glioblastoma in elderly patients. The study contains interesting results and observations; however, the authors need to answer a number of questions before the manuscript is accepted for publication:

  1. The authors need to explain briefly (1-2 sentences) at the section 2.2 the sense of evaluation of NLR, PLR and LMR in the context of circulating DNA.

Following the Reviewer’s suggestion, Section 2.2 and additionally also Section 2.3 have been supplemented with the information on the relationship between inflammatory indices and circulating DNA, and Kaplan-Meier survival curves for average cfDNA size.

Section 3. Disussion (line 233-235) has also been updated with the following information: Recently, cell-free DNA has been regarded as a damage-associated molecular pattern molecule (DAMP) that can initiate an inflammatory response in various diseases, including highly inflammatory cancers [Stortz et al. Sci 2019, Tumburu et al. Blood 2021, Dutta et al. Genes and Diseases 2023, Stejskal et al. Mol Cancer 2023].

  1. All ratios (NLR, PLR, LMR) cannot have dimensionality. (1.5*10^4 cells/ul) / (1*10^3 cells/ul) = 15 units (not cells/ul). Need to be corrected throughout the text.

Thank you for your remark. This has been corrected throughout the text.

  1. Materials and Methods:

- need to describe how CtDNA has been identified in the pool of CfDNA. How to distinguish ctDNA from cfDNA?

Identification of circulating tumour-derived cell free DNA in the pool of cfDNA is very difficult for every kind of cancer. Therefore, for several years, the profile of circulating DNA fragments (called fragmentomics) has been applied for healthy and cancer cells [Thierry AR. Cell Genomics 2023]. DNA fragments longer than 10000bp are likely to originate from necrotic cells, whereas DNA fragments

shorter than 1000 bp, particularly of 180 bp or multiples of this size, are reminiscent of the oligonucleosomal DNA ladder observed in apoptotic cells [Rostami et al. Cell Rep 2020]. According to Palande et al/ [J. Vis. Exp. 2020] 85% of circulating cfDNA fragments in cancer patients have 166 base pair (bp), 10% have 332 bp and 5% have 498 bp in length. Thierry and Sanchez [INSERM 2017. Method for Screening a Subject for a Cancer] defined a list of parametrics to distinguish cancer vs. healthy individual circulating cfDNA extracts. Overall, cancer patient circulating cfDNA shows subtle but reliable differences relative to healthy subjects. In individuals with cancer, circulating cfDNA differences include a higher number of fragments below 150 bp, as well as a lower number of fragments between 151 and 218 bp; with dinucleosome corresponding fragments peaking at 300 vs. 330 bp [Thierry Cell Genomics 2023]. Therefore, we focused on circulating cell-free DNA fragments on 50-700bp. However, whether these fragments are released from tumour cells or from activated neutrophils is another question that requires explanation. According Aube et al. {Front Immunol 2023], the size of DNA fragments released from neutrophils during NETosis is approx. 150 bp.

Section 1. Introduction has been updated with the above information.

Additionally, Section 4. Materials and Methods has been expanded to include technical details of DNA isolation and length analysis. The total circulating cell-free DNA fragments were isolated by Cell-Free AX DNA Mini Kit in accordance with the manufacturer’s protocol (A&A Biotechnology, Gdansk, Poland). Cell-free AX Kit is dedicated for isolation of cell free DNA from serum/plasma (sample volume 100 mL) and circulating tumor DNA (ctDNA). Due to high quality of isolated DNA, it can be used for a wide spectrum of applications, e.g. PCR, real-time PCR, next-generation sequencing, detection and monitoring of cancer. Then, cell-free DNA ScreenTape Kit (Agilent Technologies, Inc. Headquarters, Santa Clara, CA, USA) was used to determine accurate length (from 50 bp to 700 bp) and concentration of cfDNA at fully automated electrophoresis Cell-Agilent 4150 Tape Station (Agilent Technologies, Inc. Headquarters, Santa Clara, CA, USA).

- the authors describe in detail how the normality of a distribution can be determined, but do not specify how the data of a non-normal distribution are described. They should be described as the median 25%;75%) but not as the mean/median ± standard deviation. If you have data with non-normal distributions in your study or not, please, adjust the description of Statistical analysis.

All the statistical notes indicated by the Reviewer have been included in and under Table 1 and Table 2.

  1. Discussion:

- if ctDNA is 1% of CfDNA or less, what kind of DNA underlies the 4-fold increment of cfDNA in GBM. What is the source of this CfDNA - need to be discussed;

There are no available data on high-grade glioma, which indicates that cfDNA detection in glioma patients for clinically relevant purposes remains a challenging and complex problem. The comparative study demonstrated that glioma shed the least cfDNA into circulation compared with other cancers, even in patients with advanced disease [Gao et al. Innovation (Camb) 2022].

In turn, Jones et al. [Neurosurgery 2021] showed that cfDNA fragments released by apoptotic and/or necrotic cells typically reached a concentration 0.1-5% of total cell-free DNA in cancer patients. This means that apoptosis and/or necrosis is not the main process responsible for the increase in circulating cfDNA concentration. In our study, fragments of 50-700bp in length constituted 74.38% of total cfDNA and were 4-fold higher in GBM than HC samples, which indicates NETosis as the main cause of the high level of cfDNA fragments in GBM patients. NETs were detected in the same pathological conditions in which high concentrations of circulating cfDNA were reported, such as autoimmune diseases, inflammatory diseases, sepsis, thrombotic illnesses, and cancer. The higher neutrophil counts and their longer lifespan (activated in cancers) in patients with cancer could increase the formation of NETs and NETs by-products [Pastor et al. iScience 2022]. In health, Moss et al. [Nat Commun 2018] estimated that circulating cfDNA was derived from 55% white blood cells, 30% erythrocyte progenitors, and 10% vascular endothelial cells, accounting for 95% of cfDNA.

Discussion has been updated with the above information.

- need to discuss the possibility of the described approach for:

--- diagnosis and prognosis of GBM in younger patients;

We agree with the Reviewer that research on diagnosis and prognosis of GBM in younger patients could be extremely interesting. Unfortunately, approx. 90% of GBM primarily occur in older patients, whereas in younger patients, it usually developments from lower grade glioma [Louis DN et al. Acta Neuropathol 2016, Thakkar JP et al. Cancer Epidemiol Biomarkers Prev 2014]. In our previous retrospective study, we observed patients with low-grade glioma (1st grade n=9 aged 41.6 ± 21.9 years, 2nd grade n=32aged 46.0 ± 12.9 years) and high-grade glioma (3rd grade n=82 aged 57.3 ± 12.1 years, 4th grade n=235 aged 61.4 ± 13.1). Kaplan-Meier survival curves using cut-off values obtained from ROC curves showed a significantly higher risk of death in patients aged 63-90 than in those aged 23-63 years. In the same retrospective study, patients aged £30 years constituted a group of only 11 individuals among 358 patients with newly diagnosed glioma who had undergone an operation in Neurosurgery Centre University Hospital in Zielona Gora [Jarmuzek P et al. J Clin Med 2022]. For this reason, we decided to continue the research in high-grade glioma, which shows the highest inflammatory status.

--- diagnosis of progression and pseudoprogression of GBM (ability to differentiate);

The current aim of our study was to determine the diagnostic value of cfDNA/ctDNA for determining the risk of rapid disease recurrence after surgery procedure in newly diagnosed patients with GBM. Patients with high cfDNA levels, i.e. those predisposed to faster cancer progression, should undergo more frequent magnetic resonance imaging. We also expect that cfDNA levels may be helpful in differentiating and evaluating MRI follow-up postoperative lesions (pseudoprogression) from actual tumor regrowth (progression), and this should be the subject of further research.

--- predicting the efficacy of chemotherapy and radiotherapy.

We greatly appreciate your remark. The diagnostic value of cfDNA/ctDNA in assessing the effectiveness of adjuvant treatment, i.e. chemotherapy and radiotherapy, is the goal of our further research. We are in the process of collecting liquid biopsy/blood.

- the authors entitled their article as "Diagnostic and prognostic value of circulating DNA fragments ...", however, they did not make any clear conclusions about the possible application of the proposed approach for the diagnosis and prognosis of GBM outcome. In the penultimate paragraph of the discussion, it is necessary to summarize the possible applications of the proposed approach and all its limitations. One of the most significant limitations of this approach is its non-specificity: the amount of CfDNA can increase in other pathologies (cancers, infections and autoimmune disorders). It should be emphasized that evaluation of CfDNA/CtDNA levels and/or spectrum is only appropriate when glioblastoma has already been diagnosed by MRI or other clinically approved approach.

We agree with the Reviewer that the total cfDNA concentration is a nonspecific marker that can increase in other pathologies conditions such as infections and autoimmune disorders. Even chronic low-grade inflammation in older individuals causes an increase in total circulating cfDNA [Wawrzyniak-Gramacka et al. Nutrients 2022]. Therefore, we detailed our analysis to investigate fragments of 50-700 bp and postulated that assessment of the cfDNA/ctDNA in the peripheral blood of patients diagnosed with GBM may be an easily accessible and cheap marker of the tendency to cancer progression. Moreover, the fragments of 50-700 bp cfDNA may be important for the differentiation of "pseudoprogression" from "progression" in MRI examinations and in monitoring the effectiveness of adjuvant treatment (chemotherapy and radiotherapy) [Nørøxe et al. Oncotarget 2019].

  1. Institutional Review Board Statement and Informed Consent Statement should be corrected to meet the requirements of IJMS. The study includes samples obtained from humans!!!

Thank you for your remark. The study was performed on blood samples from the patients with newly diagnosed high-grade gliomas who had undergone an operation in Neurosurgery Centre University Hospital in Zielona Gora. On the day of admission to hospital, each patient signed a consent to stay in hospital and participate in the study. MDPI-patients-consent-form and the original document of the Bioethics Committee at Collegium Medicum University of Zielona Gora (document No. 16/2022) were submitted to Editor IJMS MDPI on February 22, 2024.

The information about the consent of the Bioethics Committee has been included in Section 4. Material and Methods (line 339-340).

Reviewer 2 Report

Comments and Suggestions for Authors

The study "Diagnostic and prognostic value of circulating DNA fragments in glioblastoma multiforme patients" by Jarmuzek et al. deals with the diagnostic value of circulating DNA (cfDNA) in glioblastoma patients. The expansion of the diagnostic panel for cancer is important and necessary. Therefore, publications on this topic are of great scientific interest.
The manuscript is very well written and follows a clear line of content. The data obtained are worthy of publication. However, the discussion of the specificity of the DNA data in relation to GBM is far too brief.
I am not entirely convinced by the experimental design of the study in this context. It shows very interesting data on leukocytes and their accumulation in GBM patients. The data collected on the serum concentration of cfDNA and its size distribution in comparison to healthy people are also extremely important. Nevertheless, the question arises as to what to do with elevated cfDNA levels in GBM patients? If I order a blood cfDNA test in a patient who has been diagnosed with a brain tumour by MRI, for example, how does this information support the diagnosis? Won't the cfDNA concentrations and size distributions be increased/changed in all brain tumours (or tumours in general)?
The cfDNA avarage size (bp) distribution shown in Figure 2 already shows that ~half of GBM patients have values in the healthy control range.
1) Please explain why the HC and GBM groups were not assigned to one or two other groups, e.g. brain metastases of other tumour entities, low-grade gliomas, but also Alsheimer's, Parkisson's or encephalitis, etc. Only in this way could the reader be convinced of the GBM specificity of the determination. Are there any specific data on individual diseases?
2) The authors report in their introduction that the cfDNA concentration is normally in the range of 100 pg/ul. Why is the average value in the control group 1290 ug/ul, ~10-fold more than in the literature? Please explain this discrepancy and also how to compare literature values.
3) Total cfDNA seems to be an important assessment criterion. Why is the total cfDNA distribution not shown in Figure 2? Showing this is mandatory for this publication.
4) The time frame for in-depth literature research is difficult due to the journal's reviewing process. Nevertheless, I think that Figures should be self-explanatory based on the legend and the M&M section, even if you are not familiar with the method. For me, the ROC curves are impossible to interpret. The meaning of the abbreviation only appears in the materials and methods chapter. There are no more details about the method. I apologise for my ignorance and at the same time request more details in this manuscript. How were these graphs created and how are they to be interpreted? For example, how do I evaluate the threshold (red number in the graph?) What do the "RR" and "CI" values stand for?
Minor:
1) In paragraph 2.2 it should read "NLR" twice instead of "NRL".

Author Response

Review 2

We greatly appreciate your time and effort dedicated to providing feedback on our manuscript and we are grateful for the insightful comments on and valuable improvements to our paper. All the suggestions helped us to evaluate our outcomes even more precisely in order to deliver improved, high quality scientific manuscript which we hope will now meet the high standards of International Journal of Molecular Sciences.

Comments and Suggestions for Authors

The study "Diagnostic and prognostic value of circulating DNA fragments in glioblastoma multiforme patients" by Jarmuzek et al. deals with the diagnostic value of circulating DNA (cfDNA) in glioblastoma patients. The expansion of the diagnostic panel for cancer is important and necessary. Therefore, publications on this topic are of great scientific interest. The manuscript is very well written and follows a clear line of content. The data obtained are worthy of publication. However, the discussion of the specificity of the DNA data in relation to GBM is far too brief. 

We greatly appreciate the Reviewer’s remark. Introduction and Discussion have been updated with the latest information concerning changes in DNA fragments in glioblastoma. So far, the elevated levels of cfDNA have been documented in malignant tumors among adults, including two studies in glioblastoma patients [Bagley et al. Clin Cancer Res 2020, Bagley et al. Neuro-Oncology Advances 2021], relative to patients with non-neoplastic diseases [Palande et al. J Vis Exp, 2020, McMahon et al. Neurosurgery, 2022]. Our earlier meta-analysis [Jarmuzek et al. Cancers 2023] of these two studies demonstrated a worse prognosis in GBM patients with a higher cfDNA (HR 2.35, 95% CI 1.27–4.36, p<0.01).

One of these studies [Bagley et al. Clin Cancer Res 2020] reported a significant difference between overall cfDNA burden in patients with glioblastoma vs. healthy controls and a worse prognosis in patients with glioblastoma with higher cfDNA concentrations. DNA fragments released by tumor cells exhibit somatic genetic alterations such as single nucleotide variants, chromosomal rearrangements, or gene copy number variations [Jones et al. Neurosurgery 2021]. However, due to the low concentration of circulating tumor DNA, the detection technology has to be highly sensitive and specific in order to distinguish ctDNA from normal leucocyte DNA.

I am not entirely convinced by the experimental design of the study in this context. It shows very interesting data on leukocytes and their accumulation in GBM patients. The data collected on the serum concentration of cfDNA and its size distribution in comparison to healthy people are also extremely important. Nevertheless, the question arises as to what to do with elevated cfDNA levels in GBM patients? If I order a blood cfDNA test in a patient who has been diagnosed with a brain tumor by MRI, for example, how does this information support the diagnosis? Won't the cfDNA concentrations and size distributions be increased/changed in all brain tumors (or tumors in general)?

We agree with Reviewer that total cfDNA concentration is a nonspecific marker. Therefore, we detailed our analysis to investigate fragments of 50-700 bp and postulated that their assessment in the peripheral blood of patients diagnosed with GBM may be an easily accessible and cheap marker of the tendency to cancer progression. Moreover, the fragments of 50-700 bp cfDNA may be important for the differentiation of "pseudoprogression" from "progression" in MRI examinations and in monitoring the effectiveness of adjuvant treatment (chemotherapy and radiotherapy) [Nørøxe et al. Oncotarget 2019].

The cfDNA average size (bp) distribution shown in Figure 2 already shows that ~half of GBM patients have values in the healthy control range.

Currently, this figure is called Figure 3. Size distribution of circulating cfDNA fragments in HC and GBM patients. We confirm that ~half of GBM patients had cfDNA lengths in a range similar to those of healthy controls. However, the concentration distribution of cfDNA and the size distribution of 50-700 bp cfDNA was more scattered in GBM samples than in HC samples, which we find very interesting. We hypothesize that necrosis may be the main cause of cfDNA concentration scattering because 50-700 bp constitute ~74% of the total cfDNA pool in glioblastoma.

1) Please explain why the HC and GBM groups were not assigned to one or two other groups, e.g. brain metastases of other tumor entities, low-grade gliomas, but also Alzheimer's, Parkinson's or encephalitis, etc. Only in this way could the reader be convinced of the GBM specificity of the determination. Are there any specific data on individual diseases?

Glioblastoma rarely metastasizes outside the brain, with a reported incidence of approximately 0.4%–2.0% [Kurdi et al. Cancer Reports 2023]. We did not demonstrate extracranial metastases in newly diagnosed glioblastoma during the follow-up period. The information has been included in Section 4.1.

In our previous retrospective study, we observed patients with low-grade glioma (1st grade n=9 aged 41.6 ± 21.9 years, 2nd grade n=32aged 46.0 ± 12.9 years) and high-grade glioma (3rd grade n=82 aged 57.3 ± 12.1 years, 4th grade n=235 aged 61.4 ± 13.1). Kaplan-Meier survival curves using cut-off values obtained from ROC curves showed a significantly higher risk of death in patients aged 63-90 than in those aged 23-63 years [Jarmuzek P et al. J Clin Med 2022]. For this reason, we decided to continue the research in high-grade glioma (glioblastoma) which shows the highest inflammatory status (the highest changes in neutrophil counts and NLR) [Jarmuzek P et al. J Clin Med 2022].

With due respect, in our study we focused on primary brain tumor – glioblastoma - which is a separate disease unit that originates from glial tissue [Louis et al. The 2021 WHO classification of tumors of the central nervous system: A summary. Neur Oncol 2021]. Admittedly, there are some interesting studies on cfDNA in neurodegenerative diseases [Pollard et al. Front Neurol 2023, Wojtkowska et al. Int J Mol Sci], encephalitis [Peng et al. Front Immunol 2019], stroke [Roth et al. Seminars in Immunopathology 2023] but the diversity in the pathogenesis of these diseases, brain location, biological material (tissue biopsy vs. liquid biopsy), type of DNA detected (nuclear cell-free DNA, tumour-derived cell free DNA and mitochondrial DNA), as well as laboratory techniques/methods used renders a reliable comparison impossible.

On balance we are extremely grateful for your comment. It is actually a very good idea to determine the length of cfDNA fragments in neurodegenerative diseases in relation to a damage-associated molecular pattern molecule (DAMP) and neuroinflammation.

2) The authors report in their introduction that the cfDNA concentration is normally in the range of 100 pg/ul. Why is the average value in the control group 1290 ug/ul, ~10-fold more than in the literature? Please explain this discrepancy and also how to compare literature values. 

The concentration of cfDNA in our healthy controls was 1290 ± 851 pg/mL. The most of GBM patients demonstrated a very high concentration of cfDNA >2000 pg/mL (77%), which could aggravate immunoinflammatory reactivity. In HC group, cfDNA >1000 pg/mL occurred in 47%, mainly in the oldest ones aged >70 years. According to Kustanovich et al. [Cancer Biol Ther 2019], cfDNA concentration is low in health (within 100 pg/mL) but it increases significantly under pathological conditions, including cancer and inflammaging >1000 pg/mL [Jylhävä et al. Aging Cell 2013, Wawrzyniak-Gramacka et al. Nutrients 2022]. Older individuals aged 80.3 ± 8.0 years: low-grade inflammation 747 ± 93 pg/mL, high-grade inflammation 966 ± 148 pg/mL [Wawrzyniak-Gramacka et al. Nutrients 2022]. However, according to our experience, the concentration changes also depend on the appropriate method, e.g. the fluorometric method lowers the values [Morawin et al. Front Public Health 2023] compared to electrophoresis [the current study]. Therefore, the assessment of cfDNA changes requires the recruitment of a reference group i.e. healthy controls of similar age.

3) Total cfDNA seems to be an important assessment criterion. Why is the total cfDNA distribution not shown in Figure 2? Showing this is mandatory for this publication.

Thank you very much for your remark. Figure 2 and Figure 3 have been included in Section 2.3. DNA analysis.

4) The time frame for in-depth literature research is difficult due to the journal's reviewing process. Nevertheless, I think that Figures should be self-explanatory based on the legend and the M&M section, even if you are not familiar with the method. For me, the ROC curves are impossible to interpret. The meaning of the abbreviation only appears in the materials and methods chapter. There are no more details about the method. I apologise for my ignorance and at the same time request more details in this manuscript. How were these graphs created and how are they to be interpreted? For example, how do I evaluate the threshold (red number in the graph?) What do the "RR" and "CI" values stand for? 

Abbreviations of ROC (receiver operating characteristic curve), AUC (area under the ROC curve), 95%CI (95% confidence interval) and RR (relative risk) have been explained in Section 4.6 Statistical analysis, and also under Figure 1 and Figure 4. Relative risk (RR) is the ratio of the risk for an event for the exposure group to the risk for the non-exposure group. 95% confidence interval (95%CI) for RR is an interval which is expected to contain the RR. In other words, we have only a 5% chance of being wrong and 95% chance that the RR belongs to the given interval.

Statistical analyses and figures were performed using R 4.2.1 software [R Core Team (2022). R: A language and environment for statistical computing. R Foundation for Statistical Computing, Vienna, Austria. URL https://www.R-project.org/

Minor:
1) In paragraph 2.2 it should read "NLR" twice instead of "NRL".

This has been corrected throughout the text.

Reviewer 3 Report

Comments and Suggestions for Authors

In this study, Jarmuzek et al. delve into the challenges in diagnosing, prognosing, and monitoring of GBM patients. Current practices involve assessing molecular markers in tumour biopsies which necessitate invasive neurosurgical procedures and pose risks to patients. Alongside limitations in imaging modalities. The author proposes circulating cell-free DNA fragments (cfDNA) as a promising biomarker for preoperative diagnosis and prognosis of GBM patients, emphasizing its potential as an early diagnostic and prognostic tool. This approach offers advantages including easy accessibility, low cost, and minimal invasiveness, enhancing GBM patient care. The study draws upon Jarmuzek et al.'s previous meta-analyses, which investigated the prognostic relevance of inflammatory immunological markers and cfDNA. These findings indicated a potentially unfavourable prognosis for GBM patients with elevated cfDNA levels. Upon reviewing, there are several areas which require improvement. Overall, the study lacks a robust connection between cfDNA and inflammatory immunological markers, which is essential for supporting the conclusions drawn. Furthermore, the data provided regarding the survival probability of GBM patients based on cfDNA size is confusing and insufficient to suggest that 201-303bp cfDNA could be considered a potential marker for therapeutic decision-making, as asserted by Jarmuzek et al. Additional feedback is provided below for the authors. Given these limitations, the study does not meet the publication standards of the International Journal of Medical Sciences in its current state and requires revisions before being considered for publication.

Major Comments

1.     There is insufficient data to highlight inflammation as a key contributor to cfDNA release. Despite the author's assertion of a high correlation between 50-700bp cfDNA and the neutrophil/lymphocyte ratio (NLR), the absence of a table or figure displaying this correlation raises questions. A mere couple of comments will not suffice. It is recommended to present this crucial data in the form of a table or figure, explicitly demonstrating the high correlation, with statistics such as Rs=0.550, p<0.001 clearly referenced.

2.     The author explicitly notes that almost all their study patients with GBM demonstrated a very high concentration of cfDNA above 1000 pg/uL, and in their HC group, cfDNA of >1000 pg/uL is observed. However, this statement is inaccurate, as data reveals that the mean total cfDNA pg/uL for the HC group is 1290, indicating an overlap between the groups. Additionally, the high standard deviation value for total cfDNA pg/uL, which stands at ±7374, further complicates the interpretation. Although a significant difference exists in the total cfDNA pg/uL between these two groups, depiction of this through graphical charts and figures would provide valuable insight.

3.     Figure 4, which examines the survival probability and number at risk for GBM patients based on two different average size bp groups, requires reassessment. Since at the median survival time (0.5 on y-axis), the 201-303 bp group exhibits a worse survival of approximately 50 days compared to the 303-416 bp group, which demonstrates a better survival of over 100 days. This conflicts with the conclusions drawn from the study's findings and analysis, and directly opposes the discussion's suggestion of considering 201-303 bp cfDNA as a potential marker for therapeutic decision-making. Particularly noteworthy is that the data lacks statistical significance (p = 0.87).

The author alleges that very high concentrations of cfDNA above 1000 pg/uL are associated with increased immunoinflammatory reactivity of neutrophils, yet there is a lack of supporting data to substantiate this claim. It is recommended to include such information in a graphical representation for clarity and authenticity of the study's findings.

4.     It may be true neutrophils are early responders in the general inflammation setting, but this may not always be true for cancers. In GBM specifically, neutrophils form a minority population, especially when compared to other immune cells. Additionally, neutrophils can exert both anti-tumoral and pro-tumoral effects. The functional effects of the NLR ratio is unclear, and its functional association to cfDNA levels, are unclear and should be addressed more formally in the discussion.

5.     In the ‘Materials and Methods’ section of this paper, it is crucial to provide detailed information regarding the timing of blood sample collection. For instance, specifying whether the samples were obtained on the day of surgical resection, immediately before skull opening, or on the day of consent and scheduling of the operation. Establishing a standardized collection timepoint is imperative, as levels of white blood cells (WBC), WBC-derived inflammatory markers, and cfDNA may vary among patients based on the timing of sample collection.

Minor Comments

1.     The study focuses on the analysis of circulating cfDNA and ctDNA fractions within the context of inflammation, proposing them as potential biomarkers for early diagnosis or prognosis in glioblastoma. However, there is a lack of supporting data or thorough explanation in the background section regarding this assertion. The author should provide more information and expand on the rationale behind this focus.

2.     Additionally, cfDNA itself cannot assess inflammation. If the study implies that high concentrations of cfDNA correlate with elevated neutrophil/lymphocyte ratio (NLR) and neutrophil levels, this claim necessitates evidence. The author should either provide evidence supporting this correlation or, if unavailable, rephrase the statement to avoid confusion.

3.     It would be beneficial to address the rationale behind using Luo et al.'s reference values for examining white blood cell count and inflammation markers in the study. Considering the p-values for data such as monocytes, platelets, and platelet-to-lymphocyte ratio (PLR) are not statistically significant. It might be advantageous to explore additional reference values alongside Luo et al.'s. This could enhance the comprehensiveness of the analysis and interpretation of inflammatory indices.

4.     It is advisable to annotate all receiver operating characteristic (ROC) curves, (Figure 1 and Figure 3) and indicate that the red dot on each plot represents the optimal threshold value. This will aid readers in understanding the significance of the plotted data points.

5.     The 50-700 bp range of cfDNA pg/uL is a crucial parameter extensively discussed and presented in both Table 2 and figures 3 in the study It is advisable to provide an explanation for why this range is important and its relevance. Currently, there is only a brief mention in the discussion section, whereas incorporating this information earlier would provide better clarity and understanding for readers.

6.     The author frequently utilizes the term "high-grade glioma," including in the introduction of the paper, stating that the study focused on patients with newly diagnosed high-grade gliomas, and not solely GBM. Moreover, the implications of the results throughout the paper suggest potential diagnostic value for clinical prognosis in patients with high-grade glioma, rather than GBM alone. This inconsistency is noteworthy, considering that the research question and aims specifically address GBM. Therefore, it would be advisable for the study to revise the title and text to specify the exact tumor types and samples under investigation.

7.     For future studies, it is recommended to explore cfDNA in the context of specific subtypes of high-grade gliomas and molecular subtypes of GBM. This approach would provide stronger evidence for the generalization of cfDNA use in clinical diagnosis and prognosis across different glioma subtypes, enhancing the applicability and precision of cfDNA-based biomarkers.

8.     For future research, the author should consider investigating cfDNA at various time points along the GBM patient pathway. Studies have indicated that cfDNA levels can fluctuate during tumor progression, postoperative recovery, and standard treatment interventions, offering potential for real-time assessment. For instance, Bagley et al. observed a rise in plasma cfDNA concentration in patients with GBM concurrently with or preceding to radiographic evidence of tumor progression. Integrating such temporal analyses would provide valuable insights into the clinical utility of cfDNA biomarkers in monitoring disease progression and treatment response (Bagley et al., https://doi.org/10.1158/1078-0432.CCR-19-2533).

Comments on the Quality of English Language

Good

Author Response

Review 3

We greatly appreciate your time and effort dedicated to providing feedback on our manuscript and we are grateful for the insightful comments on and valuable improvements to our paper. All the suggestions helped us to evaluate our outcomes even more precisely in order to deliver improved, high quality scientific manuscript which we hope will now meet the high standards of International Journal of Molecular Sciences.

Comments and Suggestions for Authors

In this study, Jarmuzek et al. delve into the challenges in diagnosing, prognosing, and monitoring of GBM patients. Current practices involve assessing molecular markers in tumour biopsies which necessitate invasive neurosurgical procedures and pose risks to patients. Alongside limitations in imaging modalities. The author proposes circulating cell-free DNA fragments (cfDNA) as a promising biomarker for preoperative diagnosis and prognosis of GBM patients, emphasizing its potential as an early diagnostic and prognostic tool. This approach offers advantages including easy accessibility, low cost, and minimal invasiveness, enhancing GBM patient care. The study draws upon Jarmuzek et al.'s previous meta-analyses, which investigated the prognostic relevance of inflammatory immunological markers and cfDNA. These findings indicated a potentially unfavourable prognosis for GBM patients with elevated cfDNA levels. Upon reviewing, there are several areas which require improvement. Overall, the study lacks a robust connection between cfDNA and inflammatory immunological markers, which is essential for supporting the conclusions drawn. Furthermore, the data provided regarding the survival probability of GBM patients based on cfDNA size is confusing and insufficient to suggest that 201-303bp cfDNA could be considered a potential marker for therapeutic decision-making, as asserted by Jarmuzek et al. Additional feedback is provided below for the authors. Given these limitations, the study does not meet the publication standards of the International Journal of Medical Sciences in its current state and requires revisions before being considered for publication.

Thank you for your comment. The following excerpt: “Current practices involve assessing molecular markers in tumour biopsies which necessitate invasive neurosurgical procedures and pose risks to patients“ has been included in Section 1.Introduction.

Major Comments

  1. There is insufficient data to highlight inflammation as a key contributor to cfDNA release. Despite the author's assertion of a high correlation between 50-700bp cfDNA and the neutrophil/lymphocyte ratio (NLR), the absence of a table or figure displaying this correlation raises questions. A mere couple of comments will not suffice. It is recommended to present this crucial data in the form of a table or figure, explicitly demonstrating the high correlation, with statistics such as Rs=0.550, p<0.001 clearly referenced.

We greatly appreciate the Reviewer’s remark. We’ve decided to remove the value of correlation as insufficient evidence of the relationship between 50-700bp cfDNA and inflammation nad/or NETosis.

  1. The author explicitly notes that almost all their study patients with GBM demonstrated a very high concentration of cfDNA above 1000 pg/uL, and in their HC group, cfDNA of >1000 pg/uL is observed. However, this statement is inaccurate, as data reveals that the mean total cfDNA pg/uL for the HC group is 1290, indicating an overlap between the groups. Additionally, the high standard deviation value for total cfDNA pg/uL, which stands at ±7374, further complicates the interpretation. Although a significant difference exists in the total cfDNA pg/uL between these two groups, depiction of this through graphical charts and figures would provide valuable insight.

In HC group, cfDNA >1000 pg/mL was recorded in 47%, mainly the oldest ones aged >70 years. According to Kustanovich et al. [Cancer Biol Ther 2019], cfDNA concentration is low in health (within 100 pg/mL) but it increases significantly under pathological conditions, including cancer and inflammaging >1000 pg/mL [Jylhävä et al. Aging Cell 2013, Wawrzyniak-Gramacka et al. Nutrients 2022]. Earlier, we observed significantly lower cfDNA concentration in low-grade inflammation (747 ± 93 pg/mL) than in high-grade inflammation group (966 ± 148 pg/mL) in older individuals aged 80.3 ± 8.0 years [Wawrzyniak-Gramacka et al. Nutrients 2022]. According to our experience, the changes in circulating cfDNA levels depend on the appropriate method, i.e. the fluorometric method lowers the values [Morawin et al. Front Public Health 2023] compared to electrophoresis in the current study. Therefore, the assessment of cfDNA changes requires the recruitment of a reference (healthy) group at similar age, and/or also requires standardization of the method.

Section 2. Results has been re-edited. Most of our GBM patients demonstrated high concentrations of cfDNA >2000 pg/mL (n=31), including n=7 very high cfDNA >10000 pg/mL, which  could potentially aggravate immunoinflammatory reactivity. Recently, cell-free DNA has been regarded as a damage-associated molecular pattern molecule (DAMP) that can initiate an inflammatory response in various diseases, including highly inflammatory cancers [Stortz et al. Sci 2019, Tumburu et al. Blood 2021, Dutta et al. Genes and Diseases 2023, Stejskal et al. Mol Cancer 2023]. This is our first observation concerning cancer but we have previously shown a significant relationship between cfDNA and inflammatory status in older non-cancer individuals [Wawrzyniak-Gramacka et al. Nutrients 2022].

The changes in cfDNA concentration have been demonstrated in Figure 2 which has been included in Section 2. Results. Individual results of GBM patients for cfDNA total concentration and average size were markedly scattered compared to HC. Could comorbidities impact cfDNA fragmentation? Probably not, because our GBM patients did not have any other diseases (acute inflammation, auto-inflammatory disease, other cancers etc.). The 50-700 bp fragments constituted ~74% of circulating total cfDNAin GBM samples (Table 2), so the remaining 26% of cfDNA could be the fragments  >700 bp originating from necrosis. However, further analysis is needed to investigate cfDNA distribution with regard to other lengths.

  1. Figure 4, which examines the survival probability and number at risk for GBM patients based on two different average size bp groups, requires reassessment. Since at the median survival time (0.5 on y-axis), the 201-303 bp group exhibits a worse survival of approximately 50 days compared to the 303-416 bp group, which demonstrates a better survival of over 100 days. This conflicts with the conclusions drawn from the study's findings and analysis, and directly opposes the discussion's suggestion of considering 201-303 bp cfDNA as a potential marker for therapeutic decision-making. Particularly noteworthy is that the data lacks statistical significance (p = 0.87).

The interpretation of Kaplan-Meyer survival curves has been supplemented in Section 2.3. DNA analysis (line 184-189): Kaplan-Meier curves intersected during the observation period, in which case, the condition of proportionality of hazards was not met and differences between groups could not be concluded. This was probably triggered by a surgical procedure that reversed the conclusion. It should be emphasized that full assessment of the prognostic value of 50-700bp fragments requires further study to identify the confounding factor on a larger sample size.

The author alleges that very high concentrations of cfDNA above 1000 pg/uL are associated with increased immunoinflammatory reactivity of neutrophils, yet there is a lack of supporting data to substantiate this claim. It is recommended to include such information in a graphical representation for clarity and authenticity of the study's findings.

Following the Reviewer’s suggestion, Section 2.3 DNA analysis has been supplemented with a graphical presentation of cfDNA (Figure 2) and with additional information.

  1. It may be true neutrophils are early responders in the general inflammation setting, but this may not always be true for cancers. In GBM specifically, neutrophils form a minority population, especially when compared to other immune cells. Additionally, neutrophils can exert both anti-tumoral and pro-tumoral effects. The functional effects of the NLR ratio is unclear, and its functional association to cfDNA levels, are unclear and should be addressed more formally in the discussion.

The heterogeneity and plasticity of tumor associated neutrophils (TANs) render them crucial in the tumor microenvironment interplay. Increasing evidence suggests a dual modulatory role of neutrophils in tumor behavior and highlights the need for a reassessment of neutrophil functions in cancer initiation and progression [Galdiero et al. J. Leukoc Biol 2018]. Tumor-derived cytokines induce the presence in the blood of immature neutrophils with immunosuppressive properties and neutrophils with an ‘aged’ phenotype that are experienced cells with an increased ability to react to inflammatory stimuli, which can thus play an anti-tumoral role [Zhang et al. Nature 2015, Androver et al. Trends Immunol 2016]. Moreover, neutrophils in the tumor tissue occur in different polarized states, i.e., N1 anti-tumoral phenotype and N2 pro-tumoral phenotype, analogous to helper T lymphocyte and monocyte polarization [Massara et al. Front Immunol 2017]. Despite detected functional differences, no definitive surface markers have been identified to differentiate N1 and N2 TANs [Fridlender et al. Cancer Cell 2009, Jarmuzek et al. Int J Mol Sci 2023].Clinical studies have shown that most glioma patients experience strong neutrophilia, and that preoperative neutrophil count is correlated with GBM grade, but the mechanism of neutrophil recruitment and their role in tumor growth is yet to be defined [Weng et al. Neurol. Res. 2018]. Earlier, we observed that among immune cells, neutrophils showed the most considerable changes, especially in patients with Grade 3 and Grade 4 tumors, whereas lymphocytes, monocytes, and platelets did not exhibit significant changes compared to reference levels or Grade 1 group [Jarmuzek et al. J Clin Med 2022]. Therefore, we focused on neutrophils and NETosis as a main source of cfDNA but we are aware that this requires further study to confirm our assumptions.

  1. In the ‘Materials and Methods’ section of this paper, it is crucial to provide detailed information regarding the timing of blood sample collection. For instance, specifying whether the samples were obtained on the day of surgical resection, immediately before skull opening, or on the day of consent and scheduling of the operation. Establishing a standardized collection timepoint is imperative, as levels of white blood cells (WBC), WBC-derived inflammatory markers, and cfDNA may vary among patients based on the timing of sample collection.

Following the Reviewer’s remark, the missing information has been included in Section 4.3. Blood samples collection. The blood samples of patients with GBM were collected from the median cubital vein in the morning between 7.00 and 9.00 on the day of admission tohospital (one day before the surgical resection) and before the introduction of steroid therapy. Blood samples were collected in S-Monovette-EDTA K2 anticoagulant tubes (Sarstedt AG & Co. KG, Nümbrecht, Germany) and immediately subjected for hematology analysis in laboratory of University Hospital in Zielona Gora.

Minor Comments

  1. The study focuses on the analysis of circulating cfDNA and ctDNA fractions within the context of inflammation, proposing them as potential biomarkers for early diagnosis or prognosis in glioblastoma. However, there is a lack of supporting data or thorough explanation in the background section regarding this assertion. The author should provide more information and expand on the rationale behind this focus.

Sections 1. Introduction and 5. Discussion have been supplemented accordingly.

  1. Additionally, cfDNA itself cannot assess inflammation. If the study implies that high concentrations of cfDNA correlate with elevated neutrophil/lymphocyte ratio (NLR) and neutrophil levels, this claim necessitates evidence. The author should either provide evidence supporting this correlation or, if unavailable, rephrase the statement to avoid confusion.

We agree with the Reviewer and decided to remove the value of correlation as insufficient evidence of the relationship between 50-700bp cfDNA and inflammation.Moreover, we have included the information about cfDNA and damage-associated molecular pattern molecule (DAMP) in Section 3. Discussion.

  1. It would be beneficial to address the rationale behind using Luo et al.'s reference values for examining white blood cell count and inflammation markers in the study. Considering the p-values for data such as monocytes, platelets, and platelet-to-lymphocyte ratio (PLR) are not statistically significant. It might be advantageous to explore additional reference values alongside Luo et al.'s. This could enhance the comprehensiveness of the analysis and interpretation of inflammatory indices.

We determined haematological parameters using the Sysmex XN-1000 and compared them to the reference intervals proposed by Luo et al. [Clin Lab 2019] for Sysmex instrument. Luo et al. showed that reverence intervals for NLR, PLR, LMR and SII differed depending on the type of instrument.

We decided to calculate all inflammatory indices to verify our previous observations/results. The retrospective study of Jarmuzek et al. [J Clin Med 2022] carried out in 358 patients showed that the grade of malignancy impacted NLR, PLR, LMR and SII. The meta-analysis of Jarmuzek et al. [Cancers et al. 2023] lead to the following conclusions: “…the high diagnostic usefulness of peripheral immune-inflammatory markers NLR and PLR over SII in the prognosis of patients with GBM. Pre-operative NLR and PLR assessment can help to evaluate disease progression, optimize treatment, or introduce anti-inflammatory agents and follow-up patients with GBM. However, further prospective studies are needed to verify the reliability of the meta-analysis performed, especially with regard to the circulating cell-free DNA”.

We have referred to our previous research in 1. Introduction and 5. Discussion.

  1. It is advisable to annotate all receiver operating characteristic (ROC) curves, (Figure 1 and Figure 3) and indicate that the red dot on each plot represents the optimal threshold value. This will aid readers in understanding the significance of the plotted data points.

Thank you very much. The suggested information has been included in the figures description.

  1. The 50-700 bp range of cfDNA pg/uL is a crucial parameter extensively discussed and presented in both Table 2 and figures 3 in the study It is advisable to provide an explanation for why this range is important and its relevance. Currently, there is only a brief mention in the discussion section, whereas incorporating this information earlier would provide better clarity and understanding for readers.

Following the Reviewer’s suggestion, the relevant information has been supplemented in Section 1. Introduction.

  1. The author frequently utilizes the term "high-grade glioma," including in the introduction of the paper, stating that the study focused on patients with newly diagnosed high-grade gliomas, and not solely GBM. Moreover, the implications of the results throughout the paper suggest potential diagnostic value for clinical prognosis in patients with high-grade glioma, rather than GBM alone. This inconsistency is noteworthy, considering that the research question and aims specifically address GBM. Therefore, it would be advisable for the study to revise the title and text to specify the exact tumor types and samples under investigation.

We apologise for the imprecision of the terms. In Section 4.1. Study population the term high-grade glioma has been replaced with glioblastoma. Nowadys, the term glioblastoma (GBM) referrs to the most aggressive primary malignant brain tumour in adults – previously high-grade glioma - grade III and IV according WHO 2016 [Louis et al. The 2016 World Health Organization classification of tumors of the central nervous system: A summary. Acta Neuropathol. 2016].

The discovery of somatic mutations in the gene encoding isocitrate dehydrogenase-1 (IDH1) initiated a new period of laboratory diagnostics and classification of tumors of the central nervous system which includes gliomas. The fifth edition of the World Health Organization Classification of Tumours of the CNS, published in 2021, established new tumour types and subtypes based on profiling genome-wide DNA methylation [Louis et al. The 2021 WHO classification of tumors of the central nervous system: A summary. Neur Oncol 2021].

Consequently. the following sentence has been corrceted: The study was carried out in patients aged 65.0 ± 11.3 years (females n=24, males n=16) with newly diagnosed glioblastoma.

  1. For future studies, it is recommended to explore cfDNA in the context of specific subtypes of high-grade gliomas and molecular subtypes of GBM. This approach would provide stronger evidence for the generalization of cfDNA use in clinical diagnosis and prognosis across different glioma subtypes, enhancing the applicability and precision of cfDNA-based biomarkers.

Thank you very much for  a very interesting idea. We collected material (BioBank - liquid biopsy and tissue biopsy) from patients with low-grade glioma and glioblastoma, and we going to continue our research on glioblastoma.

  1. For future research, the author should consider investigating cfDNA at various time points along the GBM patient pathway. Studies have indicated that cfDNA levels can fluctuate during tumor progression, postoperative recovery, and standard treatment interventions, offering potential for real-time assessment. For instance, Bagley et al. observed a rise in plasma cfDNA concentration in patients with GBM concurrently with or preceding to radiographic evidence of tumor progression. Integrating such temporal analyses would provide valuable insights into the clinical utility of cfDNA biomarkers in monitoring disease progression and treatment response (Bagley et al., https://doi.org/10.1158/1078-0432.CCR-19-2533).

The current aim of our study was to determine the diagnostic value of cfDNA for determining the risk of rapid disease recurrence after surgery procedure in newly diagnosed patients with GBM. Patients with high cfDNA levels, i.e. those predisposed to faster cancer progression, should undergo more frequent magnetic resonance imaging. We also expect that cfDNA levels may be helpful in differentiating and evaluating MRI follow-up postoperative lesions (pseudoprogression) from actual tumor regrowth (progression), and this will be the subject of our further research. We also are interested in the clinical utility of cfDNA in treatment response.

Round 2

Reviewer 1 Report

Comments and Suggestions for Authors

The authors responded to the comments and made corrections to the manuscript. I recommend that the authors supplement the discussion section with the information provided in their response to the reviewer - see the paragraphs:
--- diagnosis and prognosis of GBM in younger patients;
--- diagnosis of progression and pseudoprogression of GBM (ability to differentiate);
--- predicting the efficacy of chemotherapy and radiotherapy.

Institutional Review Board Statement (line 396) and Informed Consent Statement (line 397) should be supplemented with the information provided by the authors in their response to the reviewer.
Once these conditions are met, I recommend this manuscript for acceptance for publication.

Author Response

Response to Review 1 (Round 2)

We greatly appreciate the time spent on our manuscript revision. All of the comments motivated us to re-evaluate our outcomes in order to deliver an improved manuscript.

The authors responded to the comments and made corrections to the manuscript. I recommend that the authors supplement the discussion section with the information provided in their response to the reviewer - see the paragraphs:

- diagnosis and prognosis of GBM in younger patients;

- diagnosis of progression and pseudoprogression of GBM (ability to differentiate);

- predicting the efficacy of chemotherapy and radiotherapy.

Please forgive us for this misunderstanding. Discussion has been updated with the above information (line 252-257, line 305-308).

Institutional Review Board Statement (line 396) and Informed Consent Statement (line 397) should be supplemented with the information provided by the authors in their response to the reviewer. Once these conditions are met, I recommend this manuscript for acceptance for publication.

We attach the documents: MDPI-patients-consent-form and the original document of the Bioethics Committee at Collegium Medicum University of Zielona Gora (document No. 16/2022) were submitted to Editor IJMS MDPI on February 22, 2024.

Reviewer 3 Report

Comments and Suggestions for Authors

Thanks to the authors for revising the manuscript, however couple concerns remain in the revised manuscript.  

Major Comment #1:

In the revised paper, the author has included lines 118 to 120 on page 3, stating that ‘NLR is highly correlated with 50-700bp cfDNA (rs=0.550, p<0.001) in glioma patients, indicating that tumour-associated inflammation may be the primary contributor to the release of cfDNA into the bloodstream’.

I would like to reiterate the request for data demonstrating this correlation between the inflammation index, Neutrophil-to-Lymphocyte Ratio (NLR), and cfDNA. If there is no available data, please consider using words or phrases like "co-occurrence" or "simultaneous increased trend." For instance, the author could state, "There is a concurrent rise in NLR observed alongside elevated levels of 50-700bp cfDNA in glioblastoma patients."

Major comment # 3

The author's response is unclear. Please expand on how a surgical procedure could reverse a conclusion. Furthermore, since blood collection of patient cohort and healthy controls was conducted preoperatively, it remains uncertain how a surgical procedure would affect the samples. It would be advisable to exclude the Kaplan-Meier survival curve, particularly as it does not support the paper's claims and lacks significance. I agree with the author's suggestion to use a larger sample size for the survival analysis, and I would suggest perhaps implementing matched pairs.

Once these adjustments are implemented, I believe the paper will be ready for publication.

Author Response

Response to Review 3 (Round 2)

We greatly appreciate the time spent on our manuscript revision. All of the comments motivated us to re-evaluate our outcomes in order to deliver an improved manuscript.

Major Comment #1:

In the revised paper, the author has included lines 118 to 120 on page 3, stating that ‘NLR is highly correlated with 50-700bp cfDNA (rs=0.550, p<0.001) in glioma patients, indicating that tumour-associated inflammation may be the primary contributor to the release of cfDNA into the bloodstream’. I would like to reiterate the request for data demonstrating this correlation between the inflammation index, Neutrophil-to-Lymphocyte Ratio (NLR), and cfDNA. If there is no available data, please consider using words or phrases like "co-occurrence" or "simultaneous increased trend." For instance, the author could state, "There is a concurrent rise in NLR observed alongside elevated levels of 50-700bp cfDNA in glioblastoma patients."

Please forgive us for this misunderstanding. In 2. Results and 5. Discussion sections, we have decided to remove the value of correlation as insufficient evidence of the relationship between 50-700bp cfDNA and inflammation and/or NETosis. Section 2.2. Study white blood cell count-derived inflammation indices - have been updated and the following sentence have been used: “There is a concurrent rise in NLR observed alongside elevated levels of 50-700bp cfDNA in glioblastoma patients" (line 120-122).

Major comment # 3

The author's response is unclear. Please expand on how a surgical procedure could reverse a conclusion. Furthermore, since blood collection of patient cohort and healthy controls was conducted preoperatively, it remains uncertain how a surgical procedure would affect the samples. It would be advisable to exclude the Kaplan-Meier survival curve, particularly as it does not support the paper's claims and lacks significance. I agree with the author's suggestion to use a larger sample size for the survival analysis, and I would suggest perhaps implementing matched pairs.

We apologize profusely for unclear explanations. The important point that the Kaplan-Meier method’s “…main focus is on the entire curve of mortality rather than on the traditional clinical concern with rates at fixed periodic intervals.” If the curves are parallel, it suggests that the groups have similar survival experiences. If the curves diverge or cross, it indicates differences in survival between the groups. The log-rank test (used to find optimal cut-offs in survival data – this has been supplemented in Section 4.6. Statistical analysis) is used to test whether the difference between survival times between two groups is statistically different or not, but do not allow to test the effect of the other independent variables [Goel et al. Int J Ayurvea Res 2010, Rich et al. Otolaryngol Head Neck Surg 2010].

In our study (Figure 5), Kaplan-Meyer curves analysis applied only to GBM patients divided according to the cut-off value (log-rank test) for average cfDNA size, and showed an interesting pattern.

The results of Kaplan-Meier survival curves analysis, although insignificant, showed a higher mortality risk in GBM patients with 303-416bp cfDNA than patients with 201-303bp cfDNA fragments (Figure 5). Kaplan-Meier curves were initially parallel but crossed in the further observation period, in which case, the condition of proportionality of hazards was not met and significant differences between groups could not be detected. The survival times remained higher in patients with 201-303 bp cfDNA size. The factor disturbing the course of the curves was probably the effectivity of surgical procedure (total or subtotal resection), which reversed the relation average cfDNA size to survival times. According to meta-analysis of Han et al. [Front Oncol 2020], total or subtotal resection significantly impact survival times in glioblastoma.

Section 2.3. DNA analysis has been updated with the above information (line 183-194).
